# Tricarboxylic Acid Cycle Relationships with Non-Metabolic Processes: A Short Story with DNA Repair and Its Consequences on Cancer Therapy Resistance

**DOI:** 10.3390/ijms25169054

**Published:** 2024-08-21

**Authors:** Enol Álvarez-González, Luisa María Sierra

**Affiliations:** 1Departamento de Biología Funcional, Área de Genética, University of Oviedo, C/Julián Clavería s/n, 33006 Oviedo, Spain; enolalvglez@hotmail.com; 2Instituto Universitario de Oncología del Principado de Asturias (IUOPA), University of Oviedo, 33006 Oviedo, Spain; 3Instituto de Investigación Sanitaria del Principado de Asturias, Avda. HUCA s/n, 33011 Oviedo, Spain

**Keywords:** oncometabolites, fumarate hydratase, AlkB enzyme, chromatin remodeling, non-homologous end joining, homologous recombination repair, cancer therapy resistance, isocitrate dehydrogenase, succinate dehydrogenase, MGMT protein

## Abstract

Metabolic changes involving the tricarboxylic acid (TCA) cycle have been linked to different non-metabolic cell processes. Among them, apart from cancer and immunity, emerges the DNA damage response (DDR) and specifically DNA damage repair. The oncometabolites succinate, fumarate and 2-hydroxyglutarate (2HG) increase reactive oxygen species levels and create pseudohypoxia conditions that induce DNA damage and/or inhibit DNA repair. Additionally, by influencing DDR modulation, they establish direct relationships with DNA repair on at least four different pathways. The AlkB pathway deals with the removal of N-alkylation DNA and RNA damage that is inhibited by fumarate and 2HG. The MGMT pathway acts in the removal of O-alkylation DNA damage, and it is inhibited by the silencing of the *MGMT* gene promoter by 2HG and succinate. The other two pathways deal with the repair of double-strand breaks (DSBs) but with opposite effects: the FH pathway, which uses fumarate to help with the repair of this damage, and the chromatin remodeling pathway, in which oncometabolites inhibit its repair by impairing the homologous recombination repair (HRR) system. Since oncometabolites inhibit DNA repair, their removal from tumor cells will not always generate a positive response in cancer therapy. In fact, their presence contributes to longer survival and/or sensitization against tumor therapy in some cancer patients.

## 1. Introduction

Basic cellular metabolic processes are not identical in all cells since at least cancer cells present a distinct metabolic phenotype known as “aerobic glycolysis” [1,2]. In the last decades, research has shown the relevance of the tricarboxylic acid (TCA) cycle. Metabolic changes involving this central hub for energy metabolism, macromolecule synthesis and redox balance [3,4,5] are linked to different non-metabolic cell processes. Some of these changes are related to variations in the levels of some TCA cycle metabolites in normal cellular function [5]. However, most of them are the consequence of mutations in genes related to the TCA cycle [5,6,7,8]. In fact, the discovery of these mutations, as well as their effects, represents a turning point in the fields of cancer genetics and metabolism [7,9,10,11,12]. Moreover, they show an important connection between the TCA cycle, immunity and cancer [13,14,15,16,17,18,19].

Furthermore, in recent years, an additional link has emerged between the TCA cycle and epigenetic processes [20,21,22,23,24]. Epigenetics impacts aging [25], innate immune memory [18] and, most importantly, DNA damage repair [8,11,24,26,27,28,29,30,31,32,33]. The interaction between the TCA cycle and DNA repair might be involved in cancer therapy resistance, and this is the main subject of this review.

## 2. TCA Cycle

The TCA cycle is a metabolic process taking place in mitochondria and linking glucose oxidation with the respiratory chain. This is a specific aerobic biodegradation process that starts with the catabolism of acetyl-coenzyme A (acetyl-CoA) to produce the reduced coenzymes NADH and FADH_2_, as well as CO_2_ [34] (Figure 1).

This cycle provides cells with energy and intermediates for biosynthetic pathways [12], and it is defined as the central biochemical process in eukaryotic life [18].

Since its discovery by Krebs and Johnson in 1937 [35], many reviews have been published about the TCA cycle, several of them in the last two decades. Among them, there are nice and thorough works about energy generation and/or metabolism remodeling [36,37] and even one modifying Krebs’ original concept [38]. Several reviews were also published about the TCA cycle relationships with human inborn errors of metabolism [39], neurometabolic disorders [40], immunity and inflammation [18,41,42,43], viral infections [44] and the control of mammalian stem cell fate [45].

Most of the relationships between the TCA cycle and these non-metabolic cell processes arise from the fact that some of the cycle metabolites have additional functions outside of the TCA cycle. Among them, there are direct metabolites such as citrate, α-KG, succinate and fumarate, but also new and indirectly generated metabolites, like itaconate (Figure 2A) and 2-hydroxyglutarate, 2HG (present as two enantiomers [39]) (Figure 2B) [10,15,16,17,18,19,42,43,46,47,48]. In fact, there is quite a lot of information on the additional roles of citrate [12,13,16,17,18,42,43,49,50,51,52,53], itaconate [15,16,17,18,43,49,54,55,56], α-KG [12,57,58,59,60], 2HG [16,39,61,62,63,64,65,66,67,68,69,70,71,72,73,74,75,76,77,78,79], succinate [13,15,16,18,42,43,63,80,81,82,83] and fumarate [7,43,63,84,85,86,87] in the mentioned relationships.

There is also a relevant connection between the TCA cycle and cell proliferation and cancer [40,88,89,90,91,92,93]. It constitutes the foundation for the newly described relationship we would like to highlight: the relationship established by 2HG, succinate and fumarate with DNA repair. This interaction is established either with the epigenetic control of chromatin structure [4,11,23,25] and, therefore, indirectly with DNA damage repair [8,11,24,28,29] or directly with some DNA repair proteins [8,26,27,30,31,32,33]. This new connection constitutes one of the most important side effects of the TCA cycle metabolites, since it might provide new potential approaches and new therapeutic targets [7,12,22] in cancer treatments. Specifically, the effects of metabolites on DNA damage repair, mostly inhibitory, might be linked to the suppression of drug resistance. Consequently, this effect of metabolites is especially noteworthy due to its potential implications for overcoming the resistance commonly encountered in cancer therapies.

## 3. TCA Cycle Metabolites, Cell Proliferation and Cancer

α-KG is one of the most important TCA cycle metabolites. In addition to its role in linking this cycle with non-metabolic processes, it is the fundamental key in the relationship between cell proliferation and cancer and, therefore, with the new connection with DNA repair. This metabolite is a co-substrate for the α-KG/Fe^2+^-dependent dioxygenases (α-KGDD), a family of enzymes able to catalyze the hydroxylation of different substrates, including nucleic acids, proteins, and fatty acids, using Fe^2+^ as a cofactor [94,95]. Some of the relevant enzymes in this superfamily are (i) prolyl hydrolase domain-containing proteins (PHDs) [96], of the hypoxia-sensing system; (ii) Ten-Eleven Translocation (TET) DNA demethylases [97]; (iii) Jumonji domain-containing (JmjCs) histone-lysine demethylases [98]; and (iv) AlkB homolog proteins [99] that remove nitrogen alkylation damage from the DNA [100,101].

α-KG plays a crucial role in the hypoxia-sensing system, a mechanism that has evolved in multicellular organisms to adapt to low oxygen levels [102,103,104]. The system is based on the existence of two types of proteins. The first one is that formed by oxygen-dependent hypoxia-inducible factors (HIF-α), including the most relevant HIF-1α, which can detect and activate transcriptional responses when oxygen levels are low [57,104,105]. The second type is constituted by PHD proteins, present in normal oxygen conditions, that hydroxylate different subunits of HIF-α factors, leading to their degradation in the proteasome [106,107].

The TET 5-methylcytosine (5-mC) dioxygenases catalyze the oxidation of 5-mC to 5-hydroxymethylcytosine (5-hmC) [108,109] and then to 5-formylcytosine (5-fC) and 5-carboxylcytosine (5-cC) [110]. With these reactions, they demethylate DNA in a replication-independent process [110]. They were proposed to work in regulating DNA methylation fidelity [111]. Moreover, since their inactivation contributes to both DNA [111] and RNA hypermethylation [112], they also work on the regulation of gene expression [110,112].

The JmjC histone-lysine demethylases (KDMs) form one of the two groups of histone-lysine demethylases, which are subdivided into seven families [113,114]. Described by Tsukada et al. [98] as enzymes that depend on Fe^2+^ and α-KG for their activity, they may be modulated by metabolites [115].

The AlkB homolog proteins that are present in different organisms, including humans, remove alkyl groups from nitrogen atoms present on the nitrogen bases of the DNA and also of the RNA [99,100,101,116,117,118].

The role of α-KG is equivalent in all these proteins. Low levels of this metabolite would inactivate them, maintaining (i) stable levels of HIF-α factors, even in oxygen normal levels [106]; (ii) high levels of DNA and histone methylation [21,22,23], which is key to keeping a closed chromatin structure and inhibition of gene expression [110,119]; and (iii) high levels of DNA damage that could be the origin of mutations [120,121]. All these altered processes generate cellular conditions that might be the starting point of cell proliferation.

The same effects caused by low levels of α-KG may be achieved by the aberrant accumulation of some TCA cycle metabolites due to mutated enzymes. Since those metabolites may present pro-oncogenic activity that contributes to tumor development and progression, they are called oncometabolites. This is the case for 2HG [10], succinate [122] and fumarate [84].

2HG is present in two enantiomers: D- and L-2HG, also known as (R)- and (S)-, respectively, with IUPAC nomenclature [39,62,65]. When erroneously accumulated, they contribute to the development of different types of tumors. D-2HG accumulates in certain brain tumors, such as gliomas and glioblastomas, with gain-of-function mutations in the *IDH1* and *IDH2* genes encoding the IDH1 and IDH2 enzymes, respectively [10,62,65,123,124,125]. L-2HG accumulates in brain tumors [63] and also in kidney tumors (like renal cell carcinoma, RCC) as the consequence of a mutated L-2HGDH enzyme [61,63,64] (Figure 2B).

Succinate accumulation, due to loss-of-function mutations in genes encoding the subunits of SDH [90,126,127,128,129,130], leads to the development of various types of tumors, such as paragangliomas and pheochromocytomas [9,29,126,130,131,132,133,134], gastrointestinal stromal tumors (GISTs) [127,129,132,135,136,137,138,139], renal carcinomas [131,132,140,141], pituitary tumors [131,142,143] and others [127,131,132,134,144]. All this information has led to the classification of SDH as a tumor suppressor [131,135,141,145].

The accumulation of fumarate [7,84] due to loss-of-function mutations at the FH encoding gene [84,146,147] causes the development of hereditary leiomyomatosis and renal cell cancer (HLRCC) [84,146,148,149,150,151,152]. Additionally, it is also the origin of renal cancer (RCC) [63,153,154], leiomyoma with bizarre nuclei [155], malignant pheochromocytoma and paragangliomas [133,156,157], and even breast and bladder carcinomas [126,149], as well as neuroblastomas and gliomas [158]. Moreover, the reduction of FH activity may induce disease progression in chronic myeloid leukemia [159]. Because of its role in these carcinogenesis processes, FH was classified as a tumor suppressor [84,86,158,160].

Succinate, fumarate and 2HG work as oncometabolites because they share a chemical structure similar to that of α-KG (see Figure 1) that enables them to act as competitive inhibitors of αKGDD enzymes [22,61,161,162].

The link between 2HG and the hypoxia-response pathway is unclear due to the conflicting roles of its two enantiomers [62,63,64,65]. However, the accumulation of succinate and fumarate clearly stabilizes the hypoxia-inducible factor HIF1-α in normoxic conditions [131,145,146,158,163,164,165,166], causing pseudohypoxia. Succinate, fumarate and 2HG also inhibit, although in different degrees [63], TET DNA (L-2HG appears to be a more potent inhibitor of TET enzymes than D-2HG [65]) and JmjC KDM histone demethylases. This inhibition results in increased genomic DNA methylation and chromatin condensation, which involves the oncometabolites in the epigenetic regulation of gene expression [22,61,63,135,136,147,158,162,165,167,168,169,170]. Their accumulation in epithelial cells causes epithelial-to-mesenchymal transition (EMT), giving these cells increased invasive and migratory abilities [63,85,168,171,172,173]. They are also involved in the development and progression of renal cell cancer [63]. Finally, at least 2HG and fumarate inhibit AlkB homolog proteins [116,117,118].

Moreover, these oncometabolites present other effects that might also be important for cell proliferation, such as the impairment of glutathione (GSH) production [77], increasing reactive oxygen species (ROS) production [77,78,127] or influencing tumor immune responses by affecting dendritic cells or macrophages in tumor micro-environments [174,175]. D-2HG has been found to inhibit the activity of cytochrome c oxidase in the mitochondrial electron transport chain [65]. Furthermore, fumarate integrates immune and metabolic circuits to generate monocyte epigenetic reprogramming in the induction of trained immunity through the inhibition of KDM5 histone demethylases [81,176].

## 4. TCA Cycle, Oncometabolites and DNA Repair

As already mentioned, one of the most relevant relationships between the TCA cycle and a non-metabolic process is that established with the DNA damage response (DDR) and specifically with DNA repair mechanisms [24,26,27,28,29,30,31,32,33]. DDR is the set of mechanisms that detects DNA damage, signals its presence and promotes its repair [177,178,179,180,181,182].

The different types of lesions or adducts that modify nitrogen bases and/or the deoxyribose-phosphate backbone, commonly known as DNA damage, can be detected in all kinds of organisms [120,121]. They can be generated either as the result of endogenous cellular metabolism or as the consequence of exposure to external genotoxic agents [120,121]. The types of DNA damage include (i) monoadducts—lesions that affect one nitrogen base; (ii) cross-links—intra- and inter-strand ones—which are lesions affecting two nitrogen bases in the same strand or in opposite ones, respectively; and (iii) single- and double-strand breaks—breaks in the sugar-phosphate bonds—in one (SSB) or both strands (at the same position, DSB) of the DNA molecule [120,121].

If not removed from the DNA, all these types of DNA damage may be the source of genetic instability and mutations [120,121,178,179,180,182]. To prevent this possibility, all organisms have DNA damage response mechanisms to remove the damage or to process it to avoid its immediate danger [8,120,179]. These DDR mechanisms depend on the extent of the damage [183]. They start with (1) signaling or detection of the damage; (2) the cell response to provide time for its removal/processing, which in eukaryotes includes the control of cell cycle progression; (3) the actual removal and/or processing of the damage by DNA repair systems; and (4) cell apoptosis or cell senescence that occurs when the DNA damage cannot be removed in time [120,169,177,178,179,180,181,182,184].

Among the types of DNA damage, the most toxic or lethal ones, when not removed, are the DSBs [169,177,185,186,187], which can lead to chromosomal mutations and/or rearrangements [120,169,179]. Perhaps because of their danger, there are several mechanisms that can remove or process DSBs in a fast way [187], like homologous recombination repair (HRR), non-homologous end joining (NHEJ), alternative end joining (Alt-EJ) and single-strand annealing (SSA) [120,188,189,190,191,192,193]. The HRR system is an error-free DNA repair mechanism, whereas the others may introduce errors in the DNA when removing the DSBs [120,189,191,192]. Which mechanism should act to remove a specific DSB is not completely known. It seems to depend on several factors, such as the cell cycle stage, the 53BP1 protein and also the nature of the break ends [184,188,191,192,193]. DSBs induced by external agents might present base lesions at the break sites that can play a role in the choice of the repair mechanism [187].

It is in this context of DNA damage and DDR, or specifically DNA repair that the TCA cycle presents new and important non-metabolic effects. First, because oncometabolites induce increased levels of ROS [63,67,77,78,127,194], and ROS are a source of DNA damage, both endogenous and exogenous [177,195]. Second, because by inhibiting PHD proteins, oncometabolites induce pseudohypoxia conditions [131,145,146,158,163,164,165,166]. Hypoxia contributes to ROS induction [196] and also to a decreased expression of DNA repair genes in some tumor tissues [196,197], inhibiting homology-directed repair, mismatch repair and base excision repair pathways [196,197,198].

Third, and more importantly, oncometabolites influence DDR modulation through their effects on at least four different pathways. The first one, which we call the AlkB pathway, deals with the removal of alkylation damage from nitrogen atoms of nitrogen bases, and it is inhibited by at least fumarate and 2HG [26,27,199]. The second one, the MGMT pathway, removes alkylation damage from the oxygen atom at position 6 of guanine (O^6^-G) by direct DNA damage reversal [116,200], and it is inhibited in IDH and SDH mutated cell tumors [201,202]. The last two pathways deal with the repair of DSBs, although with opposite effects: (i) the FH pathway generates fumarate in the nucleus to help with the repair of this damage [30,31,32,33,160]; and (ii) the chromatin remodeling pathway activates the HRR system to remove DSBs, and it is inhibited by the three oncometabolites [24,28,29].

### 4.1. The AlkB Pathway

The first pathway in the relationship between the TCA cycle, oncometabolites and DNA repair is established because some repair proteins, such as AlkB and its homologs in different organisms, are α-KG–dependent dioxygenases [99] that may be inhibited by oncometabolites [26,27,199] (Figure 3).

The *Escherichia coli* inducible protein AlkB was demonstrated to work as a DNA repair enzyme on induced alkylation damage [100,101,203].

Firstly, N1-methyladenine (N1-metA) and N3-methylcytosine (N3-metC) replication-blocking lesions, generated in single-stranded and double-stranded DNA, were the only described substrates of this enzyme [100,101]. Later, more lesions were found to be repaired by this enzyme: (i) all the four possible exocyclic etheno adducts of the DNA nucleobases; (ii) other DNA monoalkyl lesions, like N^6^-methyladenine (N^6^-metA), N1-methylguanine (N1-metG), N^2^-methylguanine (N^2^-metG), N^4^-methylcytosine (N^4^-metC) and N3-methylthymine (N3-metT); and (iii) even monoalkyl RNA lesions, N1-metA, N3-metC and N1-metG [116,117,118]. The enzyme can work on single- and double-stranded substrates in DNA and RNA [117,118]. 

Although all these lesions are methylated nucleobases, AlkB seems to be capable of also removing higher-order alkyl adducts [117]. Among the DNA adduct substrates of AlkB activity, special attention has been given to exocyclic N^6^-metA, N^2^-metG and N^4^-metC since they are not deleterious, do not generate mutations and might be post-replicative markers [117,203]. However, they are removed from the DNA as other mutagenic or replication-blocking adducts. This situation suggests a potential additional biological function for AlkB protein besides DNA repair, like regulating (i) the discrimination between DNA strands, (ii) the replication start or (iii) even the gene expression through the control of these post-replicative markers [117,203].

These lesions are removed by oxidative demethylation, generating oxy-ferryl intermediates that decompose and are released as aldehydes [100,116].

The AlkB homologs in mammals show a narrower range of substrates that include etheno adducts, N1-metA, N3-metC and N3-metT DNA mono-alkyl lesions, as well as mono-alkyl RNA lesions [117,118].

The connection between AlkB homologs and oncometabolites was discovered when some *IDH*-mutant glioma patients responded to a combination of alkylating agent chemotherapy, and this outcome was linked to the inhibition of the AlkB human homologs, ALKBH repair proteins, by the oncometabolite 2HG [26,62].

Like the JmjC KDMs and TET proteins, AlkB belongs to the Fe^2+^/α-KG-dependent dioxygenases [99], and the family includes nine distinct proteins in human cells (AlkB homolog ALKBH1 to ALKBH8 and FTO) [117,118]. These enzymes were inhibited in vitro not only by D-2HG [26,27,62] but also by L-2HG [27], and the repair of alkylation DNA damage was impaired in *IDH*-mutant glioma cells [26,62].

Later, fumarate was described to also inhibit these enzymes in vitro and in vivo [199]. The possible effect of succinate on the AlkB homologs has not yet been reported, although it is expected to be found.

In this pathway, the accumulation of oncometabolites leads to the inhibition of DNA damage repair, which might be a desirable situation in chemotherapy treatments.

### 4.2. The MGMT Pathway

In eukaryotes, the *MGMT* gene encodes the O^6^-methylguanine DNA methyltransferase, or MGMT, protein present in all kinds of organisms [204]. In prokaryotes, proteins with the same function are encoded by the *ada* and *ogt* genes [205,206]. The function of MGMT proteins, also known as alkyltransferases, and even as suicide proteins, is the direct removal of alkyl groups from oxygen atoms in DNA, specifically from O^6^-guanine (O^6^-G), in a process that inactivates them [116,200,204,207]. Alkylation of O^6^-G is not a very common type of induced DNA damage. Only alkylating agents following an SN1 unimolecular substitution reaction can generate it, and at low frequencies [116,120,200,204,208]. However, it is a miscoding pre-mutagenic and pre-carcinogenic DNA lesion [116,200]. It is induced by many of the alkylating agents used in cancer chemotherapy, like temozolomide (TMZ) [200,204,207,209], and its removal from the treated cells by MGMT normally results in therapy failure. In fact, low levels of, or not detectable, MGMT activity are linked to better therapy response and longer survival [201,209,210,211].

In most cases, the lack of MGMT activity is directly associated with the methylation of CpG islands on the gene promoter [210]. Consequently, the methylation status of this gene promoter is used as a prognostic and predictive biomarker in brain tumors like glioblastoma, glioma, astrocytoma or oligodendroglioma [210,212,213,214,215]. Moreover, the relevance of MGMT activity was detected for other tumors, like gastric cancer [216], GISTs [202,217] and lung cancer [218], but not for breast cancer [219].

The influence of oncometabolites on this repair process was discovered because many of the brain tumor patients with a methylated *MGMT* promoter in tumor cells were *IDH* mutants [220,221,222], and they showed the best response to chemotherapy with alkylating agents [223,224,225,226,227]. Furthermore, some malignant SDHB-mutated pheochromocytoma and paraganglioma were associated with hypermethylation of the *MGMT* promoter and responded to TMZ [228], as well as some SDH-deficient GIST tumors [229].

The *MGMT* promoter methylation seems to be linked to a hyper-methylator phenotype, also known as CpG island methylator phenotype (CIMP) that can be established through the inhibition of TET and KDM proteins by mutations at *IDH* [230,231,232,233] or *SDH* genes [202,217,228] (Figure 3).

It is clear, then, that the accumulation of at least 2HG and succinate inhibits MGMT repair through the induction of a hyper-methylator phenotype, and this is a welcome situation in the response to chemotherapy.

### 4.3. The FH Pathway

FH is an enzyme present in all kinds of organisms, prokaryotic and eukaryotic [32,33,160,234], and in these last ones, equally distributed between cytosol and mitochondria [160].

The third pathway linking the TCA cycle and DNA repair was found when studying the role of the yeast cytosolic FH enzyme [30] (Figure 4). As a member of the DDR, this enzyme is translocated to the nucleus after the induction of DNA damage, and it helps with the recognition of DSB sites in the chromatin and with cell recovery in a function that seems to be HIF-independent [30]. In fact, FH levels increase after the induction of DNA damage and when FH is not working, the cells are sensitive to DSBs because their repair is impaired, both in yeast and human cells [30].

Since its discovery, more information has been gathered on this pathway. Cytosolic yeast FH, affected by post-translational modifications, is not active when there is no damage in the DNA [160]. Phosphorylation of its Ser 46 by PAK4 kinase retains this enzyme in the cytosol of human lung cells [235]. When DNA damage is induced, FH levels increase, and the enzyme is activated by removing these modifications or by synthesizing new molecules without modifications [160], and it moves to the nucleus, specifically to DSB sites [31,32,33,160]. At least in human cells, DNA-PK phosphorylates the Thr236 position of the translocated FH molecules, leading to their interaction with the H2A.Z histone variant [31]. This histone is present in nucleosomes at DSBs and contributes to shifting the chromatin to an open conformation [236]. FH then synthesizes fumarate from malate [31,158,160]. This locally generated fumarate inhibits KDM2B histone demethylase, enhancing the dimethylation of histone H3 lysine 36 (H3K36) [31,158,160], signaling the position of DNA damage and opening the chromatin to the necessary repair proteins [237]. In fact, in human cells, all these steps lead to the accumulation of Ku70-containing DNA-PK at DSB regions for the NHEJ system to work and help with cell survival [31,32]. These findings reveal a feedback mechanism that underlies DNA-PK regulation by chromatin-associated fumarase and an instrumental function of fumarase in regulating histone H3 methylation and DNA repair [31].

Contrary to what happens in human cells, in yeast, the repair process activated by this FH pathway is the HRR, with a relevant role of this protein in the DSB resection through its interaction with the Sae2 endonuclease [32,33]. Moreover, during DNA replication stress, the fumarate in the nucleus enhances the survival of yeast lacking Htz1p histone (H2A.Z in mammals) by increasing the methylation of histone H3 lysine 4 (H3K4) through inhibition of the corresponding histone demethylase (Jhd2p) [238]. Finally, fumarase seems to indirectly influence DDR by binding to the desulfurase Nfs1p in the mitochondria, protecting it and allowing the formation of Fe–S clusters, which are essential cofactors for DNA repair enzymes [239].

The role of FH in the response to DSBs is not restricted to eukaryotic cells since a fumarase protein involved in DNA damage response and induced by the presence of DNA damage was also found in *Bacillus subtilis* [240].

This pathway, with the accumulation of fumarate at specific DNA sites, contributes to the repair of DSBs.

### 4.4. The Chromatin Remodeling Pathway

To facilitate the access of repair proteins to the damage sites for an accurate repair of DNA, the cells need to present an open chromatin structure, which may be achieved through the work of histone modifiers [186,237,241], DNA methylation control [110] and chromatin remodelers that are recruited at the DNA lesion site [186,237,241,242].

Histone modifications play a fundamental role in chromatin structure, dynamics and functions and, among them, methylation of lysine residues in H3 and H4 and their corresponding demethylations are essential to chromatin regulation [98,114,170,243]. The JmjCs histone-lysine KDM4 (A–D) demethylase family contains the most relevant histone modifiers in the context of the epigenetic regulation of chromatin structure [98,113,169,170]. They remove methyl residues from the trimethylated histone H3 lysine 9 (H3K9me3) [113,114,169], which is necessary for maintaining an open chromatin structure [170]. H3K9me3 represents a barrier to DSB repair [169], and the enzyme removing it is then part of the DDR [169,244]. In fact, the C-terminal region of KDM4D mediates its rapid recruitment to DNA damage sites, where it is required for the efficient phosphorylation of some substrates of the DDR sensor ATM protein [244]. This recruitment depends on the poly (ADP-ribose) polymerase 1 (PARP1), which ADP-ribosylates KDM4D [244].

After the discovery of the effect of oncometabolites in chromatin structure through the inhibition of KDM demethylases [22,65,136,147,158,162,165,169,170], one question started to arise: were the oncometabolites modulating DNA damage induction and/or DNA damage cellular responses in ways not related to FH, AlkB or MGMT? [183,242,245,246]. The answer to this question constitutes the chromatin remodeling pathway in the relationship between the TCA cycle and DNA repair and was quite recently completely deciphered [24,28,29] (Figure 5).

The first insights into this pathway were provided by the reports about the following effects: (i) of hypoxia on inducing DNA damage and inhibiting DNA repair [196,197,198]; (ii) of D-2HG on inducing an impaired HRR system that rendered cells sensitive to DNA damaging agents and PARP1 inhibitors [28,247]; and (iii) of succinate and fumarate on inducing this same HRR problem, with the same consequences [29,248].

These findings uncover an unexpected connection between oncometabolites, altered DNA repair and genetic instability [28]. They were followed by results showing that the repair activity of DSBs induced by ionizing radiation was markedly reduced in *IDH* mutant cell lines [28] and in *FH* and *SDH* mutant cells [29]. Moreover, *IDH* mutant cells and those with increased levels of fumarate and succinate show an increased persistence of unrepaired DSBs, even in untreated cells, as shown with the neutral comet assay [28,158]. However, it is not clear whether 2HG, succinate or fumarate induce DNA damage in vitro [28,29,245,246] since exogenous exposures to any of them increase the levels of DSBs [24,246].

In normal cellular conditions, the induction of DSBs generates a very rapid, high and short-timed increase of H3K9me3, brought down by KDM4 demethylases [28,29]. This is the most important signal for activating the HRR system through its interaction with TIP60 acetyltransferase [249]. However, if the levels of H3K9me3 were increased because exposure to oncometabolites inhibits KDM4 demethylases, this signal is masked [24]. The lack of this signal impairs the activation of TIP60 [249]. Consequently, the recruitment of ATM protein to sites of DSBs is inhibited, which, in turn, attenuates the foci of the RAD51 and BRCA2 proteins. The impairment of these proteins, all of which are essential for normal HRR activity, inhibits this system, with a defect at the end-resection step [24].

In this chromatin remodeling pathway, the accumulation of oncometabolites inhibits the repair of DSBs. This lack of repair might be a desirable situation in radiotherapy treatments.

In addition to directly impairing the HRR system, this chromatin remodeling pathway might be linked to the inhibition of other DNA repair systems, like the nucleotide excision repair (NER), the base excision repair (BER) or the mismatch repair (MMS) systems through maintaining a closed chromatin structure. In that case, the inhibition of TET DNA demethylases [167], together with that of KDM histone demethylases, would be the key to the oncometabolite control process, although some control through maintaining hypoxia levels should not be excluded [196,197,198].

Obviously, chromatin remodeling might be achieved through other processes, some of which are also related to the TCA cycle, like the histone acetylation/deacetylation alternation, which is also used to devise chemotherapy treatments with the use of histone deacetylation inhibitors [250,251,252]; however, since the link with the TCA cycle is the acetyl-CoA metabolite [253,254,255,256] and not the oncometabolites, no more information about this process will be provided in this review.

## 5. Oncometabolites, DNA Repair and Therapy Treatment Resistance

The activities of many chemotherapy drugs and of radiotherapy depend on the generation of DNA damage that would induce apoptosis and cell death in the treated tumor cells [257,258]. It is evident, then, that functional DNA repair systems that remove the DNA damage before it kills the cells would help with tumor survival and, therefore, contribute to therapy resistance [179,224,258,259,260].

In fact, DNA repair systems have been linked to drug resistance since a long time ago [261,262,263], especially when dealing with relevant chemotherapy drugs (because they are the best available treatment for some specific tumor types), like, for instance, cisplatin or TMZ [224,257,258,264,265]. The role of DNA repair in drug resistance is so relevant from a negative point of view that there are proteins from the different repair systems that are used as biomarkers, prognostic and/or predictive, in tumor therapy [201,210,212,213,214,215]. Moreover, some processes in the repair systems are used to develop selective agents targeting specific DNA repair pathways [181,257].

As previously discussed, increased levels of oncometabolites are the origin of tumor development; therefore, a good approach to treating these tumors was to reduce such high oncometabolite levels using inhibitors of the mutant enzymes that generate them [5,63,266,267,268,269,270,271,272,273]. This type of treatment is only possible for *IDH* mutant tumors since IDH enzymes are the only ones with gain-of-function mutations [7,10,62]. However, this limitation is not a problem, considering the many tumor types due to increased 2HG levels [67]. Although the use of inhibitors provides a good response at the beginning of the treatment, after some time, some patients develop chemotherapy resistance [268,269,271,274,275,276]. The resistance is linked to (i) new additional mutations in *IDH* genes [273,277]; (ii) the already mentioned hyper-methylator phenotype of oncometabolites, which may contribute to resistance depending on the silenced genes [268,274,278]; (iii) the inhibition of ferroptosis [279,280]; or (iv) the angiogenesis-promoting phenotype of tumor-associated macrophages (TAMs), which is supposed to be activated by hypoxic tumor cell-derived oncometabolites, as demonstrated in the case of succinate [281].

In addition, high levels of oncometabolites seem to be directly related to resistance to different therapy drugs by several very specific mechanisms, as already reviewed [274]. In particular, high levels of oncometabolites may directly promote cancer treatment resistance (i) by upregulating the nuclear factor NRF2 pathway, which inhibits apoptosis and increases both antioxidant responses and the expression of efflux proteins, like in the case of fumarate through the KEAP protein [282] or succinate through hyper-succinylation [274]; (ii) by the reduction of SDH activity, which increases succinate levels and stabilizes HIF-1α protein after treatment with transforming growth factor β (TGF-β) in osteosarcoma cells [283]; (iii) by inhibiting the anti-tumor immune response, like in the case of 2HG [284]; or (iv) by promoting angiogenesis, like fumarate and 2HG [282]. Additionally, resistance to the topoisomerase II inhibitor etoposide was linked to the presence of mutated *IDH* genes but not to the accumulation of 2HG metabolite [285].

The connections between TCA cycle oncometabolites and DNA repair might be the reason for some of the therapy resistance cases. When they are present, like in the case of the locally generated fumarate, they help with the repair of induced DNA damage [30,31,32,33,286]. When the oncometabolites are removed from the cells, the repair systems are no longer inhibited [24,26,27,28,29,287,288] (Figure 6). However, the oncometabolite effects on therapy responses were not always identified in the detected responses. Some of the resistance responses were considered cases that do not respond to chemotherapy with TCA cycle enzyme inhibitors [266,289]. Other treatment responses were viewed as cases of longer survival in the presence of oncometabolites [220,224,288,289,290,291] or as a sensitization effect of oncometabolites [224,272,273,291,292,293]. None of these responses were ever related to the oncometabolite effects on DNA repair.

One of the pathways identified so far between the TCA cycle and DNA repair is that established with the MGMT protein: the oncometabolites create a hyper-methylator phenotype in tumor cells that inhibits *MGMT* gene expression [202,217,228,229,230,231,232]. In tumors treated with alkylating agents, the presence of oncometabolites inhibits the synthesis of this protein [220,221,222], and it re-sensitizes [223,288,294] the cells that were resistant to therapy due to MGMT activity [201,209,210,211,295].

Increased levels of oncometabolites also contribute to sensitization against alkylating agents via inhibition of the human ALKBH1 to ALKBH8 and FTO proteins, homologs of the AlkB protein. Since these proteins repair alkylation DNA and RNA damage, they protect against this kind of damage [288,296]. In fact, these proteins are described to be responsible for chemotherapy resistance in different tumors, not only to alkylating agents [297] but also to cisplatin [298]. Their inhibition by 2HG and fumarate sensitizes the tumors against these chemicals [26,62,288,295]. However, the depletion of FTO and ALKBH5 in epithelial ovarian tumors renders them resistant to PARP inhibitor treatment [299].

The cases of longer survival in the presence of oncometabolites and many of the sensitization ones were detected in patients with tumors identified as *IDH* mutant [224,272,288,289,290,291,292,293,300] (Figure 6). Nevertheless, these types of cases might also be expected in *SDH* and *FH* mutant tumors if they could be compared with equivalent non-mutant ones. In addition to their effect on *AlkB* genes, these cases are due to the role of oncometabolites in the inhibition of KDM proteins, which triggers the impairment of the HRR system [24]. When the chemotherapy-induced DNA damage cannot be repaired because of the oncometabolite levels, tumor cells die (by whatever pathway), and patients present with longer survival [8]. In this case, effects would be expected when cancer therapy is performed with agents inducing DSBs, such as ionizing radiation, radiation mimetic chemicals or topoisomerase II poisons, like etoposide, for instance.

It should be noted that to prevent tumor resistance to therapy, a novel strategy that combines metabolic/epigenetic alterations with immunotherapy is being used to obtain better antitumor responses [284,301,302,303].There is another effect of oncometabolites, specifically of fumarate, that might be important in drug resistance. As discussed in the FH pathway, the combination of FH activity with the NHEJ system in the nucleus contributes to the repair and/or processing of DSBs in DNA and, consequently, also to the maintenance of genomic stability and cell survival [31,32]. The role in this pathway gives FH a second activity as a tumor suppressor [32]. However, the result of this activity is opposed to the FH effects on the AlkB and chromatin remodeling pathways. In the FH pathway, a functional FH contributes to DNA damage repair through the generation of fumarate. In the AlkB and chromatin remodeling pathways, a mutated, non-functional FH inhibits DNA damage repair processes, also through the generation of fumarate. Why does fumarate induced at DSB sites not inhibit DNA repair? And why does fumarate induced in the mitochondria not help with DNA repair? The levels and localization of fumarate increase, as well as the time in the cell cycle when fumarate accumulates in both cases, which might explain these differences [32]. Furthermore, although in cultured cells, sensitization to chemotherapy with cisplatin was linked to increased levels of fumarate and malate generated outside the TCA cycle [274], we have not found any case of drug resistance or sensitization linked to this FH pathway in tumor patients, perhaps because it would not be easy to detect.

## 6. Conclusions

The central hub for energy metabolism—the TCA cycle—establishes relationships with non-metabolic processes through some of its metabolites, both direct and indirect. In addition to the clear and well-studied connections between cancer and oncometabolites and between immunity and signaling metabolites, a new relationship has emerged between oncometabolites and DNA repair that already shows its relevance in the response to DNA damage.

Relevance because, on one hand, the localized accumulation of fumarate aids in the repair, or at least with the processing, of DNA damage as lethal as DSBs. And this help occurs in normal cells exposed to DSB-inducing agents.

On the other hand, relevance because the accumulation of oncometabolites in tumor cells renders them sensitive to DSBs and also to alkylation DNA damage through the inhibition of HRR, AlkB and MGMT repair systems, respectively. These effects of oncometabolites on DNA repair might be the primary reason for their name and the basis for their effect on cancer. However, when considering tumor cells, this lack of repair could be a desirable cell trait in cases of cancer therapy, both chemo- and radiotherapy. At this moment, with the exception of MGMT inhibition, this trait is not clearly exploited because many therapy treatments are based on the use of oncometabolite inhibitors. However, if you remove the oncometabolites, you also eliminate the DNA repair inhibition. Maybe this is the reason why many cases of DNA repair restoration, through the elimination of oncometabolites, are still considered only a failure of chemotherapy with TCA cycle enzyme inhibitors.

Moreover, all the information presented until now opens the possibility of further effects of oncometabolites on other DNA repair systems, although not necessarily in a direct way. Since there are not more dioxygenases among the known repair proteins, other oncometabolite effects might be expected from their role in chromatin remodeling, as indicated. Moreover, why not expect that oncometabolites might influence the ataxia-telangiectasia-related ATR protein, which is a sensor of different types of chemically induced DNA damage?

## Figures and Tables

**Figure 1 ijms-25-09054-f001:**
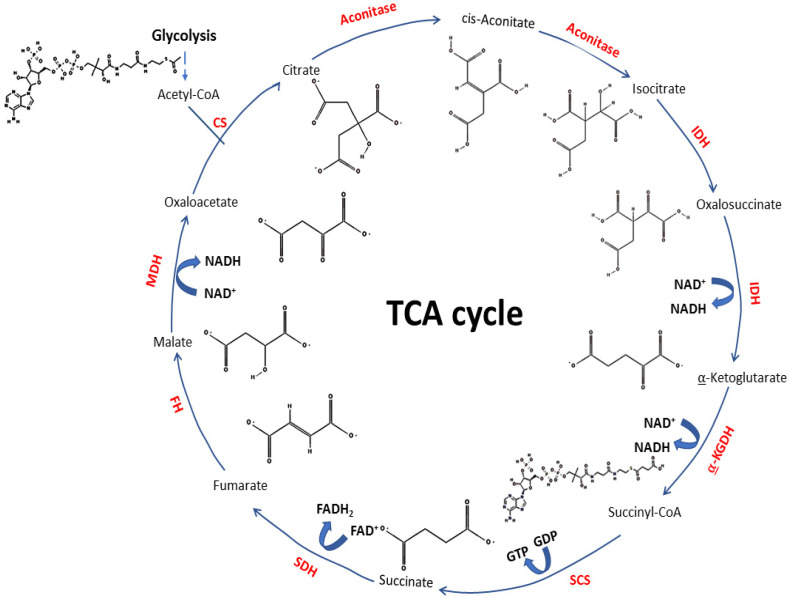
TCA cycle. Represented with the chemical formulas of the substrates/products of each reaction, indicating the catalyzing enzyme and whether the reaction generates energy-related molecules (NADH, FADH_2_ or GPT). Abbreviations: CS—Citrate Synthase (EC 2.3.3.1); Aconitase—Aconitate Hydratase (EC 4.2.1.3); IDH—Isocitrate Dehydrogenase (EC 1.1.1.42 and EC 1.1.1.41), α-KGDH—α-Ketoglutarate Dehydrogenase (EC 1.2.4.2); SCS—Succinyl-CoA Synthetase or Succinate Thiokinase (EC 6.2.1.4); SDH—Succinate Dehydrogenase (EC 1.3.5.1); FH—Fumarate Hydratase, or Fumarase (EC 4.2.1.2); MDH—Malate Dehydrogenase (EC 1.1.1.37). Formulas were obtained from PubChem (https://pubchem.ncbi.nlm.nih.gov/) (Accessed on 20 February 2024).

**Figure 2 ijms-25-09054-f002:**
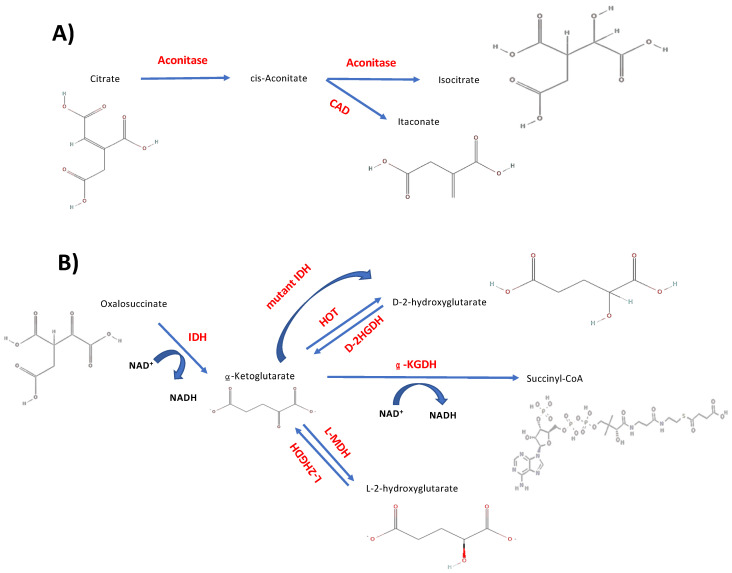
Pathways to generate TCA cycle indirect metabolites. (**A**) Itaconate, obtained from cis-Aconitate when the concentration of citrate is high. (**B**) 2-Hydroxyglutarate isomers (D- and L-2HG), generated from α-KG as waste products that can revert to their α-KG origin or as a product of gain-of-function mutations in *IDH* genes for D-2HG. Abbreviations: CAD—Cis-Aconitate Decarboxylase; IDH—Isocitrate Dehydrogenase; HOT—Hydroxyacid-Oxoacid Transhydrogenase; D-2HGDH—D-2Hydroxyglutarate Dehydrogenase; L-2HGDH—L-2Hydroxyglutarate Dehydrogenase; L-MDH—L-Malate Dehydrogenase. Formulas were obtained from PubChem (https://pubchem.ncbi.nlm.nih.gov/) (Accessed on 20 February 2024).

**Figure 3 ijms-25-09054-f003:**
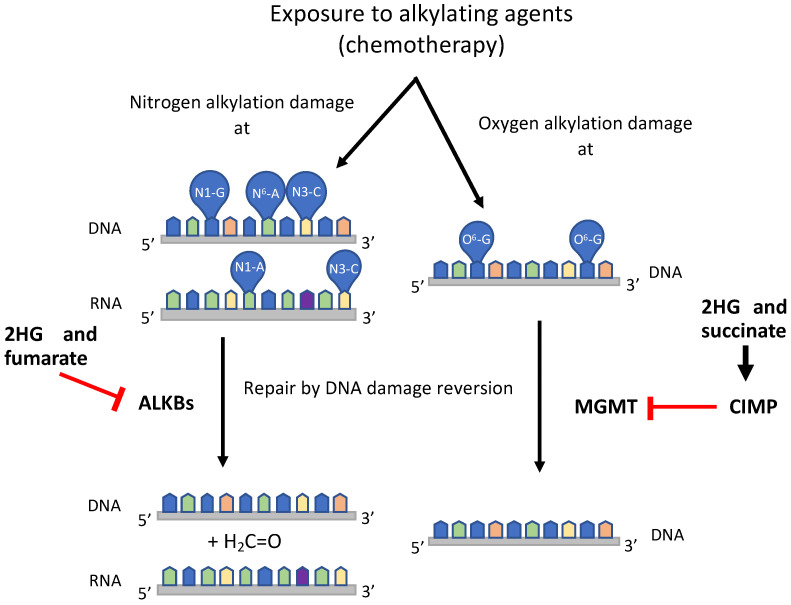
Relationships between TCA cycle metabolites and DNA repair in human cells: the AlkB pathway, following nitrogen alkylation damage, and the MGMT pathway, following oxygen alkylation damage. Human AlkB homologs remove N-alkylation damage from DNA and RNA, and 2HG and fumarate inhibit them (signaled in red). MGMT protein removes O-alkylation damage from DNA, and 2HG and succinate inhibit it through the methylation of its gene promoter (signaled in red). Abbreviations: ALKBs—human alkylated DNA repair B proteins; CIMP—CpG island methylator phenotype; 2HG—2-hydroxyglutarate; MGMT—methylguanine methyltransferase; N1-G, N3-C, N1-A—positions of ring nitrogen atoms in guanine, cytosine and adenine nucleobases; N^6^-A—nitrogen atom from the exocyclic amino group of adenine nucleobase; O^6^-G—oxygen atom from the keto exocyclic group of guanine nucleobase. H_2_C=O—formaldehyde formed as a byproduct of the reaction catalyzed by ALKBH proteins.

**Figure 4 ijms-25-09054-f004:**
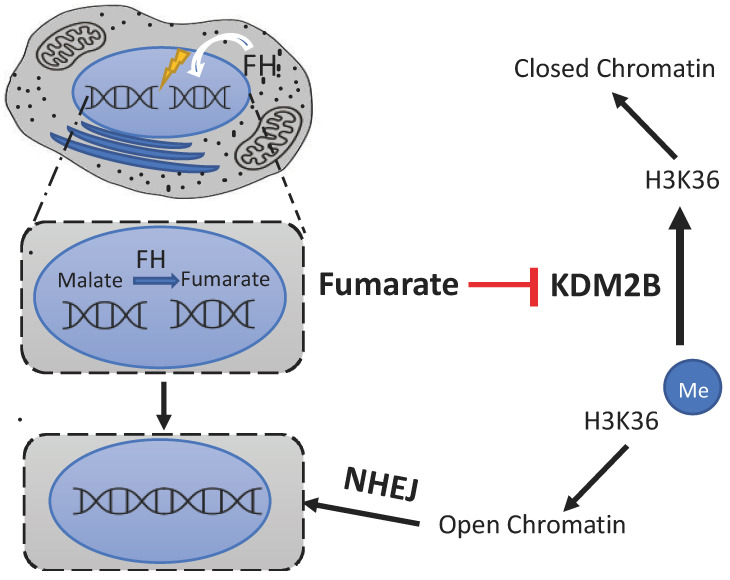
Relationships between TCA cycle metabolites and DNA repair in human cells: the FH pathway. FH protein moves from cytosol to nucleus to synthesize fumarate at DSB sites, where fumarate activates the NHEJ system. Abbreviations: DSB—double-strand break; FH—fumarate hydratase; H3K36—Lys36 in H3 histone; KDM2B—histone lysine demethylase 2B; NHEJ—non-homologous end joining system. Figure partially based on Leshets et al. [32].

**Figure 5 ijms-25-09054-f005:**
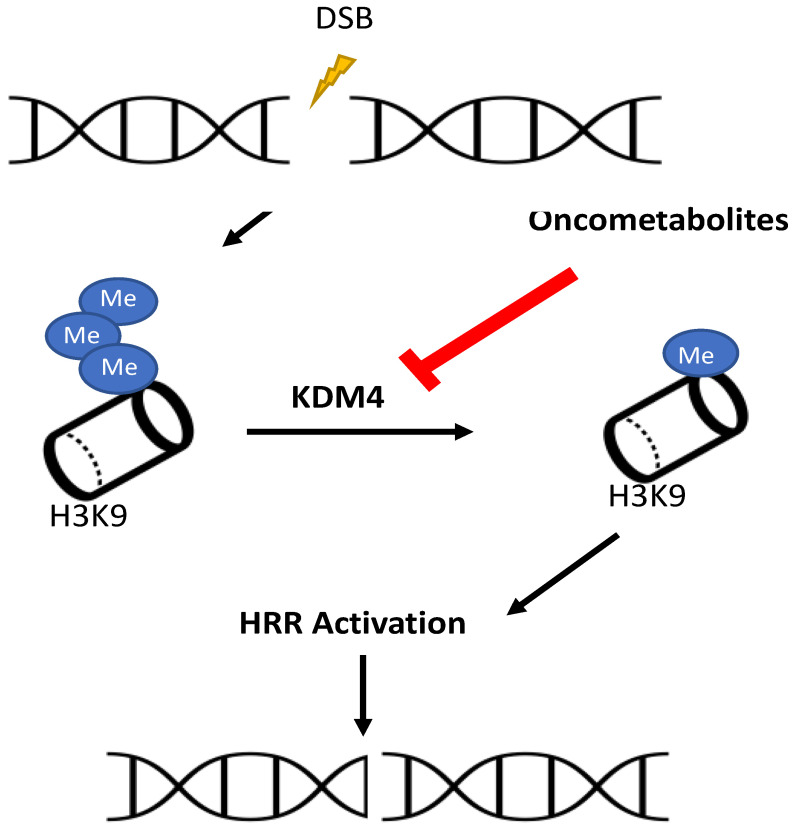
Relationships between TCA cycle metabolites and DNA repair in human cells: the chromatin remodeling pathway. Induction of DSBs increases trimethylation of H3 lysine K9. KDM4 demethylation to monomethyl H3K9 activates the HRR system. Oncometabolites inhibit KDM4 and impair repair by HRR. Abbreviations: DBSs—double-strand breaks; KDM4—histone lysine demethylase 4; H3K9—Lys9 in H3 histone; HRR—homologous recombination repair.

**Figure 6 ijms-25-09054-f006:**
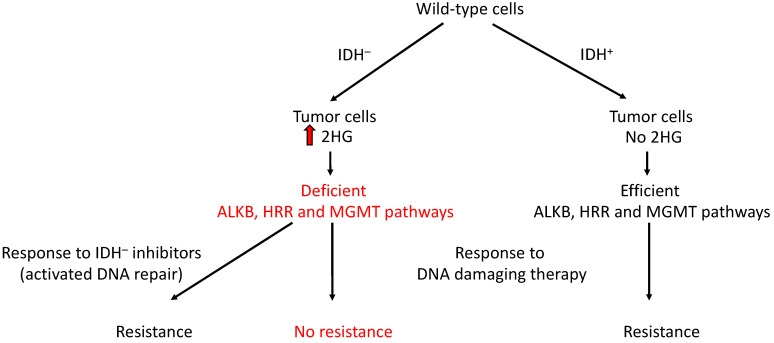
Schematic representation of the main call of this work: do not remove the oncometabolites from the tumor cells—otherwise, the inhibited DNA repair pathways would be reactivated. No resistance is achieved when repair pathways are deficient (marked in red) Abbreviations: ALKB—AlkB pathway; MGMT—methylguanine methyltransferase; HRR—homologous recombination repair pathway; 2HG—2-Hydroxyglutarate; IDH^–^—Isocitrate dehydrogenase deficient; IDH^+^—Isocitrate dehydrogenase wildtype.

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
