# Peer review of "Tricarboxylic Acid Cycle Relationships with Non-Metabolic Processes: A Short Story with DNA Repair and Its Consequences on Cancer Therapy Resistance"

_ijms, 2024, doi:10.3390/ijms25169054_

Round 1

Reviewer 1 Report (New Reviewer)

Comments and Suggestions for Authors

This manuscript deals with the relation between the non-metabolic processes of Tricarboxilic Acid Cycle (TAC) with DNA repair and the consequences on cancer therapy resistance.

The topic is of interest and the review is well organized and presented.

I would add a figure showing what described in the chapter 5 and conclusion.

There are no reference dated 2024 in the reference list.

Actually, some interesting papers have been published on this topic in 2024 and the authors could well select the most relevant of them and insert them in this manuscript.

Also, some information on the TAC cycle on immunoresonse in tumor and resistance to therapy with immune check point inhibitors could give a more wide scenario of the relevance of TAC cycle and cancer response. I understand that the authors want to focus just on DNA repair. However, immune response is a key point of tumor resistance.

Comments on the Quality of English Language

English is good

Author Response

Reviewer 2 Report (New Reviewer)

Comments and Suggestions for Authors

The manuscript describes the effects of TCA cycle on cancer-relevant processes such as DNA repair and drug resistance. It is interesting and the topic seems to be suitable for this journal. But it also has some shortcomings that need to be revised:

Acetyl-CoA fuels the TCA cycle and the vital epigenetic process of histone acetylation. While histone methylation was discussed in the manuscript, I miss the role of histone acetylation in the described processes, in particular, since several HDAC inhibitors were approved for cancer therapy and showed reasonable effects on tumor metabolism, DNA damage, and drug resistance.

Alkylating agents and cisplatin were described as DNA-damaging drugs in this manuscript. Maybe the authors can also provide relevant data on topoisomerase inhibitors, which are also salient DNA-damaging anticancer drugs?

Line 456: Please replace ´´TZM´ by ´´TMZ´´ or ´´temozolomide´´.

Figures 1 and 2: The chemical structures in these figures must be improved. They are too small and hard to read in their current form. 

The content of Figures 3 and 4 is trivial. Maybe Figures 3 and 4 can be combined into a more significant and meaningful figure.

Figure 5: Better write ´´Closed Chromatin´´.

Figure 6: Replace ´´M´´ in the blue spheres by ´´Me´´ for methylation.

Round 2

Reviewer 2 Report (New Reviewer)

Comments and Suggestions for Authors

The authors have responded properly and revised the manuscript accurately. The revised manuscript is suitable for publication now.

This manuscript is a resubmission of an earlier submission. The following is a list of the peer review reports and author responses from that submission.

Round 1

Reviewer 1 Report

Comments and Suggestions for Authors

This resubmitted version of the review did not change much relative to the original submission. Although it was shortened a bit in some places, it is still very difficult to read and understand. It still lacks focus and contains a very high level of references to reviews about related subjects. 

A major problem with the current version is still the vagueness of the writing. For example, in lines 140: 'In these cases, these metabolites ...' one needs to figure out what the two times these means. Still, it is not possible to figure out what the content of the papers is that the review refers to in many cases.

In the absence of a clear improvement, it is not really an addition to the literature and will not help fellow researchers to get a good understanding of the subject.

Comments on the Quality of English Language

Sentences are overly complicated, which makes it difficult to understand their meaning. My previous suggestion to shorten sentences has not been taken into consideration. Sentences are very long with many commas and complicated combinations of sentences in sentences.

In addition, the manuscript contains many grammatical errors (just a few examples: line 36/37 'the TCA have been linked' (instead of has), line 207 'removal/process of the damage' (instead of processing), line 292/293 'it is expected to find it to be similar' (instead of 'it is expected that similar results will be found' or something similar). 

Reviewer 2 Report

Comments and Suggestions for Authors

For the revised version of this manuscript, the authors chose not to address my major concern. 

I agree that this manuscript is not intended as a meta-analysis of genomic and transcriptomic data, and does not have to be. 

However, in several sentences the authors mention a relationship between oncometabolite presence (I assume as a result of gene expression) and therapy resistance/survival.

Currently, the authors show this by quoting several literature references, mainly in glioblastoma. Their manuscript title however suggests that the mechanisms they describe are more general.

I therefore remain of the opinion that this manuscript would benefit greatly from a brief overview of the most common aberrations of the genes described - e.g. in the TCGA set collection - and their relationship with cancer survival.

Round 2

Reviewer 2 Report

Comments and Suggestions for Authors

The authors have answered my comments. I remain of the opinion that in its current form the study is more an observation than an avenue into improved therapy. 

However, since publication might draw the attention of other clinicians and researchers, I will not object to publication.